# Assessing the Vulnerability and Risks of Adolescent Girls and Young Women in East and Southern Africa: A Preliminary Review of the Tools in Use

**DOI:** 10.3390/tropicalmed6030133

**Published:** 2021-07-15

**Authors:** Jane Ferguson, Sanyukta Mathur, Alice Armstrong

**Affiliations:** 1Independent Consultant, 1295 Tannay, Switzerland; 2HIV and AIDS Program, Population Council, Washington, DC 20008, USA; smathur@popcouncil.org; 3Adolescent and HIV/AIDS Specialist, UNICEF Eastern and Southern Africa Regional Office, Nairobi 00100, Kenya; aarmstrong@unicef.org

**Keywords:** adolescent, Acquired Immunodeficiency Syndrome, sexual reproductive health, risk assessment, risk factors, Africa, Southern

## Abstract

The sexual and reproductive health (SRH) needs of adolescent girls and young women (AGYW) aged 10–24 years remain a cause for concern in the countries of East and Southern Africa (ESA). High rates of adolescent pregnancy and HIV prevalence prevail, and prevention programmes are challenged to identify those at greatest risk. This review aimed to identify tools being used in ESA countries that support the recording of factors that make AGYW vulnerable to SRH risks and document their use. A mixed-methods approach was used to find available English language tools that had been designed to assess the vulnerability of AGYW SRH risks including literature reviews and key informant interviews with thirty-five stakeholders. Twenty-two tools were identified, and experiences of their use obtained through the interviews. All but one tool focused on HIV prevention, and most aimed at establishing eligibility for programmes, though not aligned with programme type. Analyses of the content of seventeen tools showed information collection related to behavioral, biological, and structural risk factors of HIV and other aspects of AGYWs’ lives. There was considerable diversity in the ways in which these questions were framed. Aspects of the processes involved in undertaking the risk and vulnerability assessments are presented.

## 1. Introduction

While new HIV infections among adolescent girls and young women 15–24 years old (AGYW) have declined by 19% globally between 2010 and 2017 [1], 2019 estimates indicate that in the East and Southern Africa (ESA) region 83% of new HIV infections occurred in adolescent girls aged 10–19 years [2]. Although globally the adolescent birth rate is declining, in the ESA region it was more than double the global average at 92 births per 1000 girls [3]. As much as there has been progress in terms of increasing access to HIV prevention interventions and contraceptives among adolescents in ESA [4], the uptake remains low. In response, there are calls to increase and make more effective use of external and domestic funding; improve available data and evidence to strengthen programmes; and manage the implementation of adolescent sexual and reproductive health and rights strategies at scale with quality and equity [5].

AGYW are diverse and have varying needs. Identifying who to reach with which interventions remains a challenge. Studies exist examining the determinants of adolescent sexual and reproductive health [6,7,8], conceptual models [9], and research approaches [10] that aim to better understand vulnerability to HIV among AGYW as well as the recent development of prognostic risk tools [11,12]. Organizations supporting countries to customize and target their interventions, so that they reach those AGYW most in need, recommend [13,14] programmers to assess the factors that make AGYW vulnerable and exacerbate their risk of acquiring HIV/STIs or unintended pregnancy. This review aimed to find tools being used in ESA countries that support the identification of factors that make AGYW vulnerable to sexual and reproductive health (SRH) risks and to document the experiences of using them.

## 2. Materials and Methods

A mixed-methods approach was used to identify available English language tools/methods that had been designed to assess the vulnerability of AGYW SRH risks, analyze their characteristics and experiences of using them. This included a desk review of the scientific literature from 2010–2019, conference abstracts, online resources (The database at National Center for Biotechnology Information (NCBI) at the U.S. National Library of Medicine (NLM): search strategy Mar2020 (((assessing[All Fields] AND (“risk”[MeSH Terms] OR “risk”[All Fields]) AND Adolescent[MeSH Terms] OR youth OR young OR teenage* OR child* OR adolescent* OR young adult*)) AND ((hiv[MeSH Terms] OR hiv[tw]))) OR “Sexually Transmitted Diseases, Viral”[MeSH:noexp] OR AND becoming[All Fields] AND (“pregnancy”[MeSH Terms] OR “pregnancy”[All Fields] AND ((Eastern and Southern Africa OR ESA OR Burundi OR Botswana OR Ethiopia OR Kenya OR Lesotho OR Malawi OR Mozambique OR Namibia OR Rwanda OR South Africa OR Swaziland OR Tanzania OR Uganda OR Zambia OR Zimbabwe)) not HIV/STI treatment; care; adults; published before 2010; and language other than English. Conference abstracts: Kigali—ICFP 2018, ICASA 2019; IAS 2017, 2019 (IAS abstract archive); CROI 2018, 2019; HIV and Adolescence Workshop 2018, 2020; University dissertations https://www.aau.org/current-projects/database-of-african-theses-and-dissertations-research-datad-r/, accessed on 1 March 2020; https://www.internationalafricaninstitute.org/repositories.html, accessed on 1 March 2020; websites of GFATM; IPPF; Nike & NoVo Foundations; PEPFAR/USAID; Population Council; PSI; FHI; UNICEF; UNFPA; WHO; SIDA; DFID), and a field review by means of virtual interviews with thirty-five stakeholders: tool developers, implementing partners, funders, and national programme managers. The focus of discussions with stakeholders varied although the prime interest was on the identification and subsequent use of the tools (see Table 1).

Within the different types of stakeholders, organizations, and individuals working in them were selected, based on the authors’ prior knowledge of their experience relevant to the review. Individuals were then contacted via email to explain the rationale and objectives of the review, highlight the paucity of literature on the subject and a request for an interview. Upon agreement, each person was sent questions in advance of the interview including the following core areas:The programme context in which they have they been used (e.g., screening in PMTCT service delivery points)The point in the programme development when they were used (e.g., initial planning; expansion)The purpose of the tool (e.g., seeking improvement in resource allocation and/or programme impact)The resources are required to implement tools/methods (human and financial)How the effectiveness of the tool has been assessed (dependent on its purpose)

There were most often additional questions specific to the organizational, geographic context and the interviewee. All those interviewed were requested to share any tools that they were aware of being used in ESA countries, and provide the contact details for other people who might provide information about the use of the tools. The interviews were undertaken in English, via zoom and/or local telephony for a duration of approximately one hour. Each interview was recorded with the permission of the interviewee, transcribed, and shared for revision by the individual prior to inclusion in the analysis.

The qualitative analysis of the tools was undertaken in two phases, first the analysis of those identified through the literature and web searches, and secondly of those identified from the interviews with stakeholders. A spreadsheet was prepared to record the findings across the key themes sought as part of the review (see Table 2).

An analysis was conducted of the information collected relating to AYGW risk and vulnerability which was included in 17 of the 22 tools identified. The tools excluded from the analysis were either multi-country toolkits or tools related only to health service provision.

## 3. Results

### 3.1. Overall Results and Effectiveness of the Tools

Forty resources related to the assessment of the risk and vulnerability of AGYW were identified through the desk and field reviews: twenty-two tools, twelve studies and six programme documents. This paper is limited to the tools identified that were being used in ten countries and the experiences of their use.

Except for one (set of) tools being used in pregnancy prevention projects, all the tools identified focused on HIV prevention among AGYW. There were no tools specifically related to STI prevention. It was not always clear what the specific purpose of the tools was, and when this occurred, if possible, an appraisal was made based on the discussions with the users. Details about the programme objectives and content were only broadly described. Most of the tools served to establish an individual’s eligibility for a programme and/or continuation in a programme (see Table 3).

Three studies were identified that assessed whether the tools produced the desired results:The piloting of a tool that successfully identified AGYW who were likely to discontinue school in Eswatini [15].The piloting of the Sauti vAGYW tool that reliably stratified out-of-school AGYW in Tanzania, based on individual and structural HIV risks [16].The development of tool(s) used to gain an in-depth understanding of adolescents’ needs, barriers, and motivations to use contraceptives, summarized in a report on the resulting programme design in Ethiopia [17].

There was no available documentation on the actual use of tools or their impact, apart from one report on the characteristics of AGYW recruited in the early implementation of DREAMS [18] in two countries (which suggested that the screening tools being used were not identifying the AGYW most at risk of HIV) [19].

Despite the limited information about the effectiveness of the tools being used, the people using the tools were unanimous in voicing their opinions about the *utility* of the tools for targeting their programmes because the programmes were addressing perceived needs that they thought were not always easy to identify.

### 3.2. Content of the Tools

In the 17 tools analyzed, it was possible to examine the information that was being collected about individual risk and vulnerability factors affecting the AGYW (see Table 4). At least 50% (i.e., 9 of 17) of these tools raised questions about:Behavioral risk factors: sexual activity; transactional sex; multiple partnersBiological risk factors: HIV status; ever/current pregnancyStructural risk factors: experiences of abuse and GBV; schoolingHousehold: parental presence/support.

Table 4 summarizes the categories and topics included in the tools as they relate to the available international guidance for assessing AGYW risk and vulnerability to HIV [20]. The italicized text are additional to those categories/topics in this guidance document and were found in the tools.

The number of questions posed varied, from 5 to 73, as did the style, which included questionnaires, checklists, scoring matrices, and conversation guides with probes. There was considerable diversity in the ways in which the questions were framed (see Table 5), and a range of questions on other aspects of AGYW’s lives, e.g., personal situation (see Table 4).

### 3.3. Processes in Undertaking the Risk and Vulnerability Assessments

The tools were most often administered by mentors—young women who were selected based on pre-defined criteria. Sometimes the tool administration involved teachers, social workers, and in two instances, committees with mixed membership, one of which included adolescents. How the information was recorded (and used) and the amount of interpretation required on the part of the person administering the assessments also varied.

In view of the sensitivity of questions related to sexuality and the assumed likelihood of false responses by the AGYW, interviewees emphasized the importance of adequately training the people administering the tools. Provided that there was adequate training, most people did not perceive sensitivities around sex to be challenging. In the two tools that included self-assessments, implementers thought that the responses were more honest, and noted their surprise that AGYW acknowledged transactional sex and instances of gender-based violence (GBV).

The burden of maintaining the records of the assessments was one of the challenges highlighted by several tool users. Most implementers spoke of the volume of paper, and the inconvenience of storage and subsequent reuse. This was particularly a problem when other tools were also being used in the programme, e.g., screening for health service interventions.

Procedures for the verification of the results included: ad hoc participation of supervisors in assessments; regular supervision and training; spot checks; consultations with communities and/or other professionals in the communities; and reference to health and education records. Community validation was raised several times by stakeholders as an indication of how concerned community members were to have the needs of their AGYW addressed. This concern increased the motivation of those carrying out the assessments to be accurate and fair. Unfortunately, it was not possible to confirm that such procedures could confirm the validity and replicability of the information collected through the tools.

### 3.4. Varying Priorities and Differing Needs for Information

Most of the tools were being used in projects funded by the President’s Emergency Plan for AIDS Relief (PEPFAR DREAMS project) or the Global Fund to Fight AIDS, Tuberculosis, and Malaria (GFATM) that were being implemented in specific geographical locations, usually selected by the size of the AGYW population, HIV incidence and/or pregnancy rates, and determined through national/district information management systems and/or representative surveys. The benefits of using of such population-specific aggregate data for strategic and programme planning was mentioned by several government stakeholders, although they also referred to frequent problems with the lack of age and sex disaggregation and sample sizes.

Other examples of varying programme contexts included: one tool used by a research team to establish eligibility for an intervention trial; an NGO used another to predict the likelihood of leaving school; tools from two different countries were being used in the context of appraising adolescents’ situations within household vulnerability assessments, one by an NGO and another by a government. Differences in the tools were likely related to not only to the purpose (Table 3) but also to the objectives and resources of the organizations promoting the tools, the directions given to the tool developers, and the competencies of the users. There was little information about how most of the tools were actually developed, although there was mention of influence or adaptation of The Girl Roster developed by the Population Council [21] and approaches recommended in DREAMS project guidance.

It was reported that there were often other tools in use in countries, for example, government vulnerability assessment tools also being used by different ministries, and screening tools, some of which were required by the government (e.g., HIV testing). The latter reportedly sometimes interfered with the timely completion of their own vulnerability/risk assessment tools due to the replication of questions and procedures.

Although there were indications that in some instances the vulnerability assessments included in this review, aligned with those undertaken by governments, several government stakeholders and other implementers expressed that the fragmentation and duplication of tools was a major concern.

One government official responsible for adolescent health, stressed that the efforts being undertaken at all levels by the government aimed to identify *underserved* populations and strategies to reach them more effectively, to improve overall coverage, rather than a priori targeting vulnerable sub-populations. This emphasizes the need to take into consideration different perspectives, for example, those of organizations seeking to identify AGYW who may be left out of services intended for all and those of organizations aiming to serve those thought to be particularly at risk and requiring specific interventions.

## 4. Discussion

This preliminary review of AGYW vulnerability and risk assessment tools shows commonalities in terms of content, approaches, and challenges, but also considerable variations. A common topic included in the tools analyzed, considered sexual activity as a risk factor. Although, perhaps more relevant for intervention design than programme recruitment, some stakeholders noted the need for a more nuanced understanding of AGYW sexual socialization and motivations for engaging in sexual activity. Some of the literature speaks of the value accorded by AGYW to preserving and managing their sexual relationships, and that this may supersede considerations of HIV prevention [22]. In contexts where early childbearing within or outside marriage/union may be socially accepted, or even encouraged, early pregnancy is likely to be intended and wanted (e.g., to formalize partnerships), which is never the case for HIV/STIs. Vulnerability and risk for HIV is therefore not always the same as vulnerability and risk for pregnancy.

Uncovering the desire to avoid, delay, space, or limit childbearing was central to the formative research undertaken in the ‘inquiry’ phase of programmes aiming to improve contraceptive uptake in three countries [17]. However, in only five of seventeen tools reviewed for their content were there enquiries about AGYWs’ general aspirations for the future, let alone their reproductive aspirations.

We sought to understand how the communities were involved in the application of the tools, conscious of how communities act on the challenges and needs they face, and how their perceptions are important for the design and delivery of interventions [23] for AGYW’s programmes. In our review, we learned how the perspectives of the community could also create problems, if, for example, there is pressure to enroll certain AGYWs. The opinions of community members can reinforce discrimination and negative stereotypes, particularly about the behaviors and aspirations of AGYW who seem to challenge traditional norms. The existence of a tool, with clear criteria and procedures to follow, can help counteract pressures for favoritism and provide opportunities to exchange views about social norms.

While most of the tools we identified were related to HIV prevention programming, from the discussions focusing on pregnancy prevention, we noted another perspective on the need to target and the nature of vulnerability. On the assumption that most adolescents will be sexually active at some time and are therefore, at risk of unintended pregnancy, and that since all adolescents are to some extent vulnerable, some stakeholders expressed the opinion that all adolescents require basic knowledge and access to services. This suggests a possible reluctance on the part of governments to use individual assessments to target specific subpopulations of AGYW and limit their eligibility to specific interventions. It may be that some preventive interventions are considered to be universal, i.e., applicable across the AGYW population irrespective of risk. Holmes et al. speak of such difficulties in balancing the broader goal of achieving universal health coverage with the HIV dynamic targeting approaches being recommended [24].

This review was not able to examine the content and use of the tools related to the specifics of the programmes they were used in. However, the purpose of the tool is likely linked to a specific programme or intervention, be it universal, selective (raised risk for poor outcomes), or indicated (already demonstrating risk) [25]. For example, establishing eligibility is crucial to determine the suitability of some biomedical interventions e.g., PrEP and eligibility may also be important in the case of costly interventions (e.g., education subsidies). Similarly, if the intervention is single or part of a comprehensive programme for which multiple interventions are needed to achieve outcomes, the purpose and nature of the tool will likely vary.

The prevalence of determinants of AGYW vulnerability and the health burden varies by the geographic context as does aspects of programme delivery, such the accessibility of their delivery platforms for AGYW ‘in need’. Geography thus is also likely to have a bearing on the purpose and content of tools. As previously indicated, while many tools identified and analyzed were being implemented in districts determined through aggregate data, there were examples of other processes being undertaken, e.g., HIV hotspot mapping and consultations with communities, NGOs and local councils.

There were several limitations in our methodology. Although every effort was made to conduct a systematic and inclusive review, it is possible that there are other tools and methods in use that were not identified. In a few cases, it was not possible to interview the people who had used the tools. We were not able, in all instances, to obtain instructions for use of the tools nor the complete tool. The objective of the review was to review experiences of using the tools, but no effort was made to explore the perspectives of the AGYW who had been assessed. However, several youth focused NGOs were contacted to provide and discuss the tools that they use, although no responses were received. Finally, the stakeholders’ perspectives and experiences of their tools may not be comparable to their experiences with other tools or different contexts.

However, with increased interest in the acceleration, effectiveness and efficiency of programmes designed to meet the HIV and SRH needs of AGYW, perhaps it is time to consider the development of a standard menu of content for possible inclusion in vulnerability and risk assessment tools, with a suggested repository of questions that could be aligned with government tools and data across sectors. Attention to the aims and contexts of programmes for which tools might be used will be important for the questions to be appropriate. Caution is needed to avoid the use of tools that enquire about risk and vulnerability factors without an understanding of context and the diversity of locally constructed experiences and the developmental flux of AGYWs’ development [26].

Countries can expect to benefit from global initiatives to standardize the definition and measurement of core indicators of adolescent health and age disaggregation [27,28]. These will provide greater consistency and validity to aggregate data about vulnerability and risks. The need to build capacity in the understanding and use of data, including those from programmes at community level, is also crucial [29].

During this exercise we learned little about efforts to routinely engage young people in the creation of tools to reflect upon their own circumstances and risk. Meaningful youth engagement is essential for all phases of programming: to better understand the challenges and risks that young people face and their reactions to them; their aspirations, and the opportunities and support that they need [30]. A greater involvement of young people in development of self-assessment tools, so scarce in our review, could assist stakeholders, depending on the context, to obtain more accurate information on AGYW experiences [31].

## 5. Conclusions

It is undeniable that AGYW are vulnerable to SRH risks in ESA countries with major impacts on individual and public health and well-being. This preliminary review of tools to assess their vulnerability and risks demonstrates that a range of stakeholders are earnestly engaging in processes to appraise these risks and vulnerability in order to better target prevention activities. How valid and reliable these assessments of vulnerability and risk are is not always clear and requires further attention as do the links to subsequent action, including especially programme efficiency and effectiveness. Exploring, through operational research, how the use of such tools can contribute will be crucial for replicating, adapting and scaling-up programmes that meet the needs of AGYW. An allied necessity is further investment to strengthen national information systems that facilitate disaggregated data collection, analyses, and their rapid use by a range of stakeholders.

## Figures and Tables

**Table 1 tropicalmed-06-00133-t001:** Type, focus, and number of interviews of stakeholders.

Stakeholder Function	Focus of Enquiry	No.
Tool developers	(i) availability of instruments (ii) availability of results of testing/use (iii) expectations about when and why the tools should be used (iv) resources (technical, financial) needed to use tools	3
Funders, technical supporters of programmes	(i) which tools and approaches are recommended to assess risk and vulnerability and why (ii) how capacity is built to use the tools	15
Researchers	(i) which tools/methods have they used and would recommend for programme staff and why (ii) names of people to speak to who have used tools for purposes other than research	4
Users of tools	(i) purpose of the tool (ii) experiences with its use (iii) resources needed (iv) recommendations	13
		35

**Table 2 tropicalmed-06-00133-t002:** Themes of interest in the analyses of identified tools and interviews.

Country of implementation
Tool developer
Tool administrator
Target population(s)
Selection criteria of geographic area for implementation of tool
Community involvement in development and application of the tool
Timing of tool use during programming
Resources needed for tool development and implementation
Processes for verifying results of assessments
Effectiveness of tool
Challenges encountered in the use of tool.

**Table 3 tropicalmed-06-00133-t003:** Purpose of tools identified in scoping review.

Purpose	Number
Establish beneficiary eligibility/continuation for programmes	10
Establish need for additional support/interventions	4
Record of eligibility (risk) criteria at service planning or delivery	2
Beneficiary self-assessment	1
Provide ‘baseline’ information for use in monitoring & evaluation	1
Enumeration of AGYW at risk in community	1
Service delivery monitoring	1
Market segmentation to differentiate strategies for service delivery and/or product use	1
Routine or clinical enquiry for service provision/referral	1
Total	22

**Table 4 tropicalmed-06-00133-t004:** Types of information collected in vulnerability and risk assessment tools (n = 17).

Categories of Risk Factors	Topics in Tools	No. of tools
Behavioral	*Sexually active*	11
	Transactional sex	9
	Multiple partnerships	9
	*Substance use (incl. alcohol)*	8
	*Use/non-use of HIV/pregnancy prevention practices*	7
	Age-disparate sex	5
	Gaps in knowledge (prevention)	3
Biological	*Ever/current pregnancy*	12
	*HIV status*	9
	High viral load of male partners (knowledge of partner HIV status)	7
	STI/RTIs (AGYW or partner)	6
	*Has child*	5
	Low prevalence MC (knowledge about partner MC)	2
Structural	(low) school attendance	15
	Child sexual/physical abuse	10
	GBV	10
	Married	5
	Barriers to service use	3
	Labour migration (AGYW or partner)	2
*Household*	*House head characteristics & support*	9
	*Family characteristics (numbers, illness)*	8
	*Food availability*	7
	Orphanhood	6
	*Responsibilities for family*	3
	*Characteristics of dwelling*	2
*Personal situation*	*Health status (incl. mental)*	8
	*Employment status*	7
	*Aspirations for the future*	5
	*Social relations/activities*	4
	*Sufficiency of income*	4
	*Disabilities*	4

**Table 5 tropicalmed-06-00133-t005:** Illustrative examples of the diversity in question framing across risk assessment tools.

Multiple Partnerships		
Country	Questions	Response types
A	How many sexual partners have you had in the last 12 months?	Open-ended
	(Asked in relation to 3 most recent sexual partners)	
	Is this person same age, younger or older than you?	Close-ended, categorical
	Is this partner circumcised?	Close-ended, categorical
	Do you know this partner’s HIV status?	Close-ended, dichotomous
	How often did/do you use a condom with this partner?	Close-ended, categorical
B	During the past year have you had more than one sexual partner?	Close-ended, dichotomous
	Do you think, or know, that your sexual partner(s) has other sexual partner(s)?	Close-ended, dichotomous
C	Have you ever had sex with more than one sexual partner in the last 12 months?	Close-ended, dichotomous
	Have you ever had sex with more than one partner in a day?	Close-ended, dichotomous
	Have you ever had sex with more than one partner in a week?	Close-ended, dichotomous
		Scoring of fixed response choices. Rating of Scores: 0-No risk, 1-Low risk, 2-Moderate risk, 3-High risk
House head characteristics		
Country	Questions	Response types
A	Who is the head of your household?	Close-ended, categorical
	How old is the head of the household?	Open-ended
	Is your mother/father alive?	Close-ended, categorical
	Is any of your parent/guardian chronically ill (including HIV)?	Close-ended, categorical
B	Are you living with your mother and father or with other family members?	Close-ended, dichotomous
	Who else is living in the house with you?	Open-ended
	How many people live in the house with you?	Open-ended
	Tell me a bit more about your living arrangements?	Open-ended
	[If she is not living with father/mother] Does your father/mother live elsewhere?	Close-ended, categorical
	Tell me a bit about your household circumstances. Who pays the bills in your household or who is working?	Open-ended
	Is someone getting a grant in your house?	Close-ended, categorical
	How are the finances going in your household?	Open-ended
	Are you working?	Close-ended, dichotomous
C	Do you have to provide anything for your family members like food, fees, medical care, clothing etc. (if yes, specify)	Close-ended, dichotomous
	Is there an adult person that you seek emotional and or financial support from?	Close-ended, dichotomous
	Who is the head of the family?	Close-ended, categorical
	Age of head of family	Close-ended, categorical
		Scoring of fixed response choices. Rating of Scores: 0-No risk, 1-Low risk, 2-Moderate risk, 3-High risk

## Data Availability

Publicly available tools identified in the review can be viewed in UNICEF AGYW Programming & Implementation Repository. http://www.childrenandaids.org/agyw-repository#:~:text=The%20AGYW%20Programming%20%26%20Implementation%20Repository,in%20Eastern%20and%20Southern%20Africa, accessed on 12 July 2021.

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
