# Peer review of "Assessing the Vulnerability and Risks of Adolescent Girls and Young Women in East and Southern Africa: A Preliminary Review of the Tools in Use"

_tropicalmed, 2021, doi:10.3390/tropicalmed6030133_

Round 1

Reviewer 1 Report

OVERVIEW

Thanks for this interesting paper. It’s useful to see reviews such as this about toolkits being currently used in the field, and especially with this particular focus (AGYW). It is well written, though at times (to me) felt a little like a technical report for an INGO rather than an academic paper. This is not a bad thing in itself of course, but could impact on the reader.

I do have some comments (see below). I hope they are useful.

INTRODUCTION

This provides a useful background, cites recent literature, and provides suitable justification for your review.

MATERIALS AND METHODS

This section gives details of how you selected the toolkits to review. Also, I note that formal ethical approval for the review was not required. But, as you did interview 35 respondents, there should be information about their role and where they work, how you recruited them, what you told them about the review and what would happen to the data, and broadly the questions. You were after all asking them to use up some of their valuable time to talk with you. 

I would recommend moving ‘limitations’ to nearer the end (after the discussion?).

RESULTS

This provides some useful information about the tools you included in the review. It would be interesting to see more detail about geographical context, and differences between countries.

One concern I have is that a review such as this is inevitably fragmented, given that the 22 tools you selected differ in a number of ways, clearly serving different purposes and collecting different types of data. You do partly address this in section 3.4, but the discussion on this could be expanded.

At the beginning of page 4 (para starting line 111) you say there are three studies assessing whether the tools produced the required results; but there are four bullet points. Should the last bullet point be a separate paragraph?

It’s interesting to see the types of information being collected, and section 3.3 is perhaps the most useful here, around the administration and verification processes of the tools.

The final paragraph in section 3.4 is important, though your conclusions a little subjective. It could depend on context. I would recommend rewording.

DISCUSSION

This raises a number of interesting points, especially around community involvement (or not) in developing the tools, the need for a degree of standardisation, and cultural issues in the context of traditional norms and the nature of tools targeting AGYW.

I would suggest expanding the discussion around whether the particular priorities of a tool (first flagged in the previous section, 3.4) has an impact on its usefulness for AGYW programming. This would be valuable.

I would recommend adding a short conclusion section at the end to draw the paper together and make your own recommendations more explicit, and with suggestions for further research.

Reviewer recommendations

  1. Move ‘limitations’ to nearer the end. 
  2. Add a little more detail about the interviewees and the interviewing process (especially consent – though I appreciate this will not be as rigorous as for a formal research study).
  3. Consider rewording the final paragraph of section 3.4.
  4. Consider expanding the discussion around geographical context, and the impact of a tool’s priority on the added value for AGYW.
  5. Consider expanding the discussion on key differences between the tools, why this is so, and the implications.
  6. Add a conclusions section.

Reviewer 2 Report

I consider it important to evaluate the existing tools for approaching and knowing about HIV and sexuality, especially young women. This review carried out explains well how these tools are and puts their analysis in a critical way. For me the community validation is the most important and this article reinforces it, and have a standart menu can´t invalidation this. I agree with the publication of the same and I have no request to modify the text to be carried out.

Reviewer 3 Report

Thanks for the opportunity to see this document review!

It is interesting, was well conducted, and has good writing.

Before indicating publication, I ask the authors to inform (clearly) which databases were used for the 2010-2019 document review (page 2, lines 55 to 58). This information must be included in the manuscript, indicating transparency and enabling reproduction.

Congratulations to the authors for the review!

Round 2

Reviewer 1 Report

Thank you for this revised version of the paper and for addressing my recommendations. I have no further comments and, from my perspective, I'm happy to accept the paper in its current form.